# MutAtlas: A PDB-Wide Energy-Guided Atlas of Protein Mutation Effects

Ruihan Guo [* 1]    Chaoran Cheng [* 1 2]    Zhanghan Ni [1]    Neil He [1]    Bangji Yang [1]    Ge Liu [1 2]

## Abstract

Protein mutation effect prediction is fundamental to protein engineering and disease variant interpretation, yet experimentally measured mutation data remain accurate but extremely sparse. To provide scalable supplementary mutation signals, we construct a PDB-wide mutation augmentation dataset that exhaustively enumerates single-site substitutions on experimentally resolved protein structures and aligns mutation signals from physics-based energy models, protein language models, and inverse folding models. Large-scale analysis under a unified mutation preference representation reveals substantial differences in the consistency, concentration, and substitution patterns of mutation distributions across models, indicating that disagreement is pervasive and reflects conflicting inductive biases rather than random noise. Motivated by these observations, we propose an unsupervised multi-source mutation preference distillation framework that learns from relative mutation preferences while explicitly modeling cross-source disagreement. Without using any experimental mutation labels during training, our approach achieves the best overall performance among the evaluated zero-shot baselines and naive multi-source fusion strategies on ProteinGym. We release the dataset and evaluation pipeline to support reproducible studies of protein mutation effects.

[1] Siebel School of Computing and Data Science, University of Illinois Urbana-Champaign, Urbana, IL, USA  [2] DOE Center for Advanced Bioenergy and Bioproducts, University of Illinois Urbana-Champaign, Urbana, IL, USA . Correspondence to: Ruihan Guo <ruihang6@illinois.edu>, Ge Liu <geliu@illinois.edu>.

*Proceedings of the 43rd International Conference on Machine Learning*, Seoul, South Korea. PMLR 306, 2026. Copyright 2026 by the author(s).

## 1. Introduction

Predicting the effects of amino-acid substitutions is central to protein engineering, therapeutic design, and disease variant interpretation. However, experimentally measured mutation effects, such as deep mutational scanning data (Riesselman et al., 2018), remain accurate but sparse, covering only limited proteins, sites, and functional contexts (Notin et al., 2023). This data scarcity limits the scalability and generalizability of mutation effect models.

Computational tools offer a scalable alternative. Physics-based energy functions, protein language models, and inverse folding models can generate mutation-level signals across diverse proteins. Yet these predictions are typically produced in isolation, and the field lacks a systematic, structure-aligned augmentation resource spanning the Protein Data Bank. We address this gap by constructing a PDB-wide dataset that exhaustively enumerates single-site substitutions on experimentally resolved structures and aligns multiple computational mutation signals at the residue level. This large-scale augmentation further exposes a key challenge: supervision from different modeling paradigms is heterogeneous and poorly calibrated.

This heterogeneity arises from fundamental differences in how mutation signals are generated. Physics-based methods such as FoldX estimate mutation-induced energy changes based on explicit force-field models, providing physically interpretable signals that primarily reflect local structural stability and packing constraints (Schymkowitz et al., 2005; Kellogg et al., 2011; Delgado et al., 2025). In contrast, evolution- and language-model-based approaches, including masked protein language models (Rives et al., 2021) and MSA-based models (Rao et al., 2021), capture statistical preferences shaped by evolutionary history (Riesselman et al., 2018; Notin et al., 2022; Hopf et al., 2017; Laine et al., 2019). These models produce probabilistic scores that reflect sequence compatibility rather than physical energetics, and have demonstrated strong performance in ranking mutations and estimating relative preferences (Luo et al., 2021; Notin et al., 2022; Meier et al., 2021; Guo et al., 2024). Structure-conditioned and inverse folding models (Hsu et al., 2022; Dauparas et al., 2022) further incorporate backbone geometry, but remain constrained by the distribution of available structures and training data. Recent

protein design frameworks have begun to incorporate computational feedback to improve inverse-folding-based generation objectives (Wang et al., 2026), further highlighting the importance of how computational signals are defined and combined. Importantly, these approaches do not provide noisy variants of a common underlying signal; instead, they encode fundamentally different inductive biases and operate on incompatible semantic scales.

Despite this heterogeneity, most mutation effect learning frameworks still use computational signals in simplified forms. A common formulation treats mutation scores as absolute regression targets or converts them into pairwise preferences, often centered on wild-type-to-mutant comparisons under a single reference source. This formulation becomes fragile in large-scale computational augmentation settings, where mutation sources differ in scale, semantics, and inductive bias.

It has two key limitations. First, WT-centered supervision induces a star-shaped comparison structure: each mutant is compared with the wild type, while alternative substitutions at the same residue position are rarely compared with one another. This overlooks the inherently comparative nature of protein engineering decisions, where candidate substitutions are often evaluated relative to other candidates at the same site (Shroff et al., 2020). Second, extending such supervision to multiple heterogeneous sources assumes that their induced preferences are mutually compatible. In practice, energy models, language models, and inverse folding models may favor mutations for different reasons. Directly merging their outputs can collapse informative disagreement into an artificial consensus and obscure the relative relationships among candidate substitutions.

To compare heterogeneous mutation signals without assuming a shared absolute scale, we use a unified position-wise mutation preference representation that preserves relative substitution preferences and supports cross-source analysis. Large-scale analysis of this mutation preference atlas shows that cross-source disagreement is pervasive and often reflects conflicting inductive biases rather than random noise.

Building on this observation, we make three contributions. First, we construct a PDB-wide mutation augmentation dataset that exhaustively covers single-site substitutions on experimentally resolved structures and unifies physics-based, sequence-based, and inverse-folding-based mutation signals. Second, we systematically analyze mutation distributions across sources, showing substantial differences in their substitution patterns, concentration, and cross-model consistency. Third, we propose an unsupervised multi-source mutation preference distillation framework that learns from relative preferences while explicitly modeling disagreement, improving generalization on Pro-teinGym without using experimental mutation labels. We will release the dataset, code, and evaluation pipeline at `https://github.com/guoruihan/MutAtlas`.

## 2. Related Work

### 2.1. Mutation Effect Modeling for Protein Engineering

Modeling mutation effects on protein stability and function is central to protein engineering. Computational approaches broadly include physics-based energy models and machine-learning-based predictors. Energy-function-based methods estimate mutation-induced stability changes, often expressed as $\Delta\Delta G$, by evaluating empirical or semi-empirical force fields on protein structures. Tools such as FoldX and Rosetta decompose stability into interpretable physical terms, including van der Waals interactions, electrostatics, solvation, and entropy-related effects (Guerois et al., 2002; Schymkowitz et al., 2005; Alford et al., 2017; Dehouck et al., 2009).

Machine-learning-based methods provide complementary mutation signals from sequence or structure. Sequence-based approaches, including protein language models such as ESM2 and alignment-based models such as MSA Transformer, use large-scale sequence corpora or multiple sequence alignments to score mutations through likelihood-based or coevolutionary comparisons (Rives et al., 2021; Meier et al., 2021; Rao et al., 2021). Structure-conditioned models, including inverse folding methods such as ESM-IF and ProteinMPNN, incorporate backbone geometry to estimate amino-acid compatibility under a fixed structural context (Hsu et al., 2022; Dauparas et al., 2022). These heterogeneous sources have shown strong performance across mutation benchmarks (Notin et al., 2023), but they encode different physical, evolutionary, and structural assumptions, motivating systematic comparison and multi-source integration.

### 2.2. Learning from Multiple Supervision Sources

Learning from multiple supervision sources appears in multi-teacher distillation, ensemble learning, weak supervision, noisy-label learning, and crowdsourced annotation (Hinton et al., 2015; Ratner et al., 2017; You et al., 2017). Common approaches aggregate signals through averaging, voting, or weighted combinations and train a model to match the resulting consensus (Dawid & Skene, 1979; Rodrigues & Pereira, 2018). These methods implicitly assume that sources provide noisy but compatible observations of a shared target. However, heterogeneous sources may encode distinct objectives or inductive biases, making disagreement systematic rather than purely stochastic (Han et al., 2018). In such cases, naive consensus can obscure informative structure, motivating methods that explicitly account for disagreement

among semantically different supervision sources.

# 3. Data Construction and Mutation Representation

We construct a structure-grounded dataset of single-site protein mutations and represent mutation signals from different models in a unified, residue-level format. The dataset is built on physics-based energy evaluation of mutations on experimentally resolved protein structures. All mutation signals are aligned and expressed as per-residue mutation vectors, providing a consistent data representation for downstream analysis and learning. The resulting dataset comprises 15,900,903 residue positions spanning 86,479 protein chains. For each residue position, FoldX evaluates all 19 possible amino acid substitutions, yielding a total of over 300 million single-site mutation evaluations. In terms of structural coverage, the number of protein chains included in our dataset exceeds that of the largest experimentally measured deep mutational scanning (DMS) collections by approximately two orders of magnitude. An overview of the mutation signal construction and unified representation is illustrated in Figure 1.

## 3.1. FoldX-Based Mutation Augmentation

We take protein chains with experimentally resolved three-dimensional structures from the Protein Data Bank (PDB) (Berman et al., 2003) as the starting point. For each PDB chain, we extract residues that are present in the resolved structure and belong to the standard 20-amino-acid alphabet. These residues are concatenated in sequence order to form a structure-aligned target sequence associated with the given protein structure.

Based on this structure-aligned sequence, we perform exhaustive single-site mutation augmentation using FoldX (Schymkowitz et al., 2005). FoldX is a physics-inspired empirical energy function that evaluates the effect of point mutations by explicitly mutating side chains, performing local structural repacking, and computing the resulting change in folding free energy between the wild-type and mutant structures. For each residue position, FoldX enumerates all possible amino acid substitutions and outputs mutation-level energy differences associated with specific residue positions and substitutions. These physics-based mutation signals form the core data substrate of the constructed dataset. Details of the FoldX mutation evaluation protocol are provided in Appendix C.

## 3.2. Alignment and Unified Mutation Representation

**Alignment of mutation signals.** In addition to FoldX-based energy evaluation, we collect mutation signals from sequence-based and structure-conditioned models. Specifi-

cally, we obtain position-wise mutation logits from a masked protein language model (ESM2) (Lin et al., 2023) and from an inverse folding model (ESM-IF) (Hsu et al., 2022).

To ensure consistent alignment across models, all mutation signals are computed on the same structure-aligned residue sequence. Throughout this work, we restrict the mutation space to the canonical set of 20 standard amino acids. This guarantees that mutation signals from FoldX, ESM2, and ESM-IF correspond to the same residue positions and the same mutation space.

**Unified mutation representation.** FoldX produces mutation-specific energy differences, whereas ESM2 and ESM-IF naturally output position-wise amino acid logits. To express all mutation signals in a common format, we convert FoldX energy differences into position-wise mutation logits.

Let $E(i, a)$ denote the FoldX-predicted energy of mutating residue position $i$ to amino acid $a$, and let $a_{\mathrm{WT}}$ denote the wild-type amino acid at that position. We define the FoldX mutation logit as

$$\ell_{i,a}^{\mathrm{FoldX}} = -\big(E(i, a) - E(i, a_{\mathrm{WT}})\big). \tag{1}$$

This transformation maps FoldX energy differences to a per-position, per-amino-acid logit representation. After this conversion, all mutation signals are expressed as 20-dimensional vectors associated with individual residue positions.

We define the fundamental data unit in this work as a single residue position paired with a 20-dimensional mutation vector. This unified representation standardizes mutation data across models and enables systematic downstream analysis and learning.

# 4. Mutation Landscape Analysis Across Models

We perform a systematic comparison of mutation-scoring behaviors across ESM2, ESM-IF, and FoldX under the unified mutation preference representation introduced in Section 3. For each residue position, the output of each source is represented as a 20-dimensional preference distribution over the standard amino-acid alphabet. This representation allows us to compare sources within a shared mutation space without assuming that their raw scores have a common scale or semantic meaning.

Our goal is not to determine which source is universally correct, but to characterize how different modeling paradigms shape mutation preferences. We therefore analyze three complementary aspects of the mutation landscape: (i) *substitution preference patterns*, (ii) *native distributional sharp-*

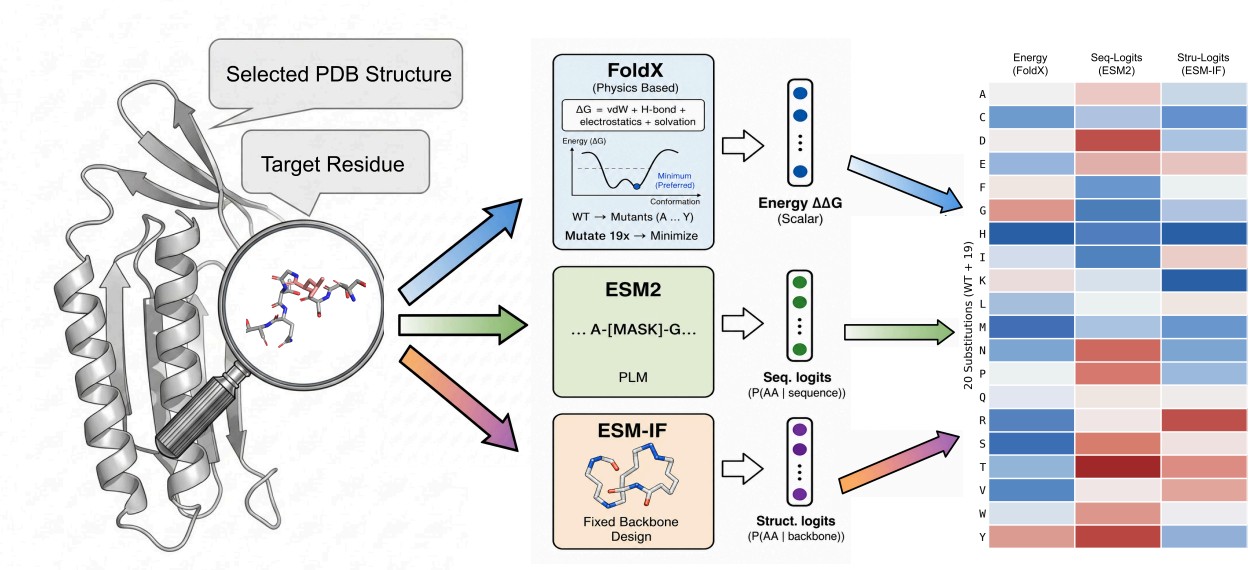

*Figure 1.* Overview of mutation signal construction and representation. For a residue selected from an experimentally resolved protein structure, mutation signals are obtained from three complementary sources: physics-based energy evaluation using FoldX, sequence-based prediction using a masked protein language model (ESM2), and structure-conditioned inverse folding using ESM-IF. All signals are aligned to the same residue position and expressed as 20-dimensional mutation logits over the standard amino acid alphabet, forming a unified per-residue mutation representation.

*ness*, and (iii) *cross-method consistency at the decision and ranking levels*. All analyses are conducted across all residue positions from all structures.

### 4.1. Substitution Preference Patterns

**Wild-type Self-preference.**    We first examine whether different sources tend to preserve the wild-type (WT) amino acid at the decision level. For each WT residue type, we compute the fraction of positions where the WT amino acid is selected as the Top-1 mutation. As shown in Fig. 2, ESM2 and ESM-IF strongly favor WT across residue types, exhibiting conservative mutation behavior. In contrast, FoldX more frequently promotes non-WT substitutions, indicating a less conservative mutation landscape.

This difference should not be interpreted simply as a difference in confidence or accuracy. Rather, WT preservation reflects the objective and inductive bias of each source. Sequence- and structure-conditioned neural models are biased toward residues compatible with natural protein sequences or fixed backbone contexts, whereas FoldX evaluates local energetic favorability and therefore need not preserve the native residue as strongly. Additional WT self-

preference analyses, including WT probability mass and relaxed Top-3 WT inclusion, show the same overall trend and are provided in Appendix B.1.

**WT-to-Mutant Substitution Landscapes.**    We next characterize broader WT-to-Mutant substitution behavior at the decision level. As shown in Fig. 3, ESM2 and ESM-IF induce structured substitution landscapes with strong WT-to-WT dominance. FoldX, in contrast, exhibits a flatter landscape, with weaker WT preservation and broader preference over alternative mutant residues.

Source-specific behavior also persists beyond WT identity. When aggregating Top-1 mutant selections across all positions, FoldX shows a stronger global tendency toward small and conformationally flexible residues, whereas ESM2 and ESM-IF display distinct mutant preference profiles. Thus, the differences among sources are not only caused by how strongly they preserve the original residue, but also by their intrinsic preferences over the amino-acid alphabet. Full score-level and decision-level substitution matrices, together with global mutant preference statistics, are provided in Appendix B.

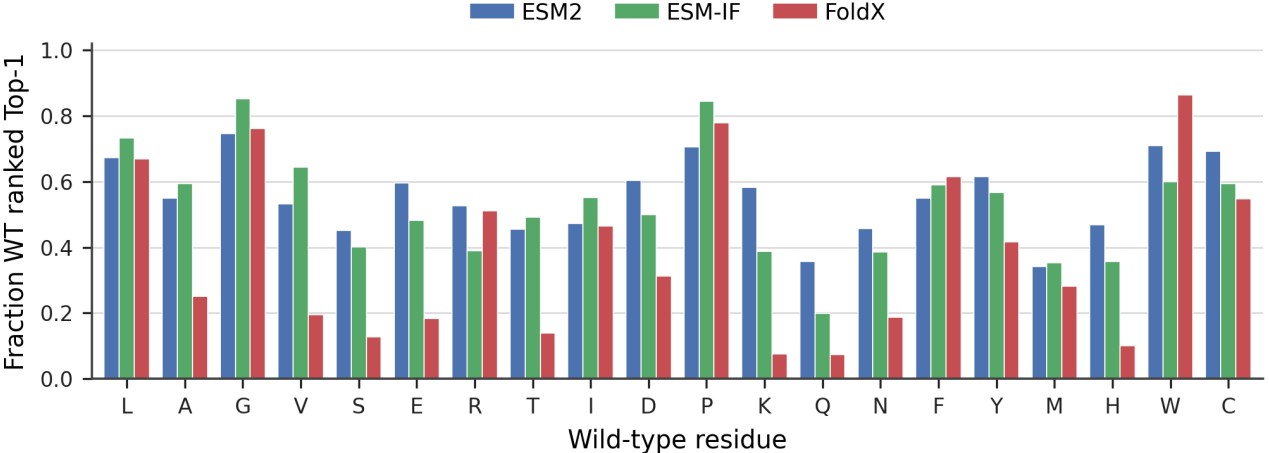

*Figure 2.* **WT self-preference at the decision level.** Fraction of positions where the WT amino acid is selected as the Top-1 mutation, grouped by WT residue type. ESM2 and ESM-IF exhibit strong WT preservation, whereas FoldX more frequently promotes substitutions.

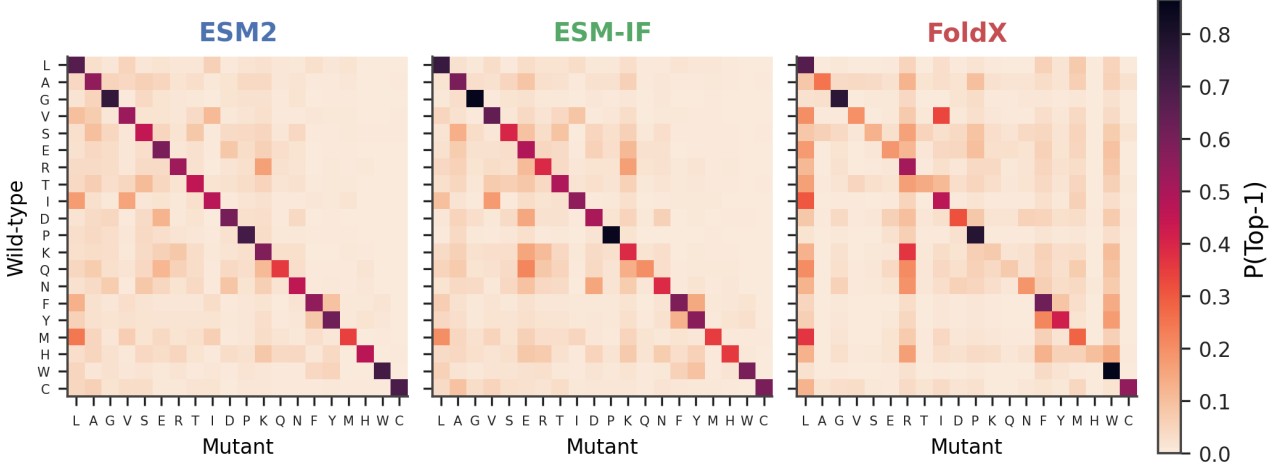

*Figure 3.* **WT-to-Mutant substitution behavior at the decision level.** FoldX shows weaker WT preservation and broader substitution preferences compared to ESM2 and ESM-IF. Full score-level and decision-level landscapes are reported in Appendix B.

## 4.2. Native Distributional Sharpness

We next analyze how sharply each source expresses its mutation preferences. Here, native distributional sharpness refers to the concentration of each source's own preference distribution before any cross-source calibration or fusion.

As shown in Fig. 4, FoldX exhibits consistently higher per-position entropy, indicating broader and less sharply peaked mutation distributions. In contrast, ESM2 and ESM-IF show lower entropy at a subset of positions, reflecting more decisive local preferences. The same pattern is also observed at the decision-margin level: Top-1–Top-2 logit gap distributions show that FoldX generally produces smaller local margins, whereas ESM2 and ESM-IF produce larger margins at a subset of positions. Therefore, the difference in

distributional sharpness is not specific to entropy, but also appears in the separation between leading candidate substitutions. Supporting decision-margin analyses are provided in Appendix B.4.

## 4.3. Cross-Method Decision and Ranking Consistency

We evaluate cross-method agreement at both the decision and ranking levels. Agreement on the Top-1 mutation is generally low across all method pairs (Fig. 5, left), indicating that different sources often propose distinct optimal substitutions at the same residue position.

To obtain a scale-invariant comparison of full mutation preferences, we further analyze per-position mutation rankings using Spearman rank correlation. ESM2 and ESM-IF ex-

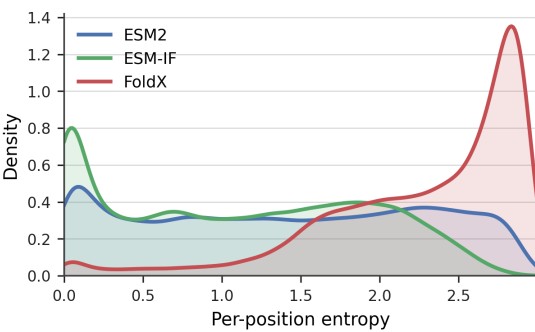

*Figure 4.* **Per-position uncertainty in mutation space.** FoldX exhibits substantially higher entropy than ESM2 and ESM-IF, indicating broader and less confident mutation preference distributions.

hibit higher ranking similarity, whereas FoldX shows substantially lower correlation with both neural sources (Fig. 5, right). This pattern suggests that FoldX provides a complementary ranking signal rather than a noisy version of the same preference structure.

The observed disagreement is not merely an artifact of strict Top-1 comparison. When the criterion is relaxed to Top-$k$ overlap with $k = 3$ and $k = 5$, agreement remains limited across method pairs. This suggests that different sources often prioritize different regions of mutation space rather than only swapping the top-ranked amino acid. Supporting Top-$k$ overlap analyses are provided in Appendix B.5.

Finally, within-source uncertainty does not fully explain cross-source disagreement. Although high-entropy positions are more likely to show disagreement, we also observe many low-entropy positions with high cross-method disagreement. These cases are particularly informative: individual sources can make sharp predictions while disagreeing with one another. This supports our interpretation that disagreement reflects conflicting inductive biases rather than simply noisy or uncertain predictions. Supporting entropy–disagreement analyses are provided in Appendix B.6.

**Section 4 Summary.** Across substitution preferences, distributional sharpness, and ranking structure, ESM2, ESM-IF, and FoldX encode complementary and structurally distinct mutation signals. Their differences are systematic: they persist beyond WT self-preference, remain visible under relaxed agreement criteria, and cannot be explained solely by within-source uncertainty. These findings indicate that cross-source disagreement reflects divergent inductive biases rather than numerical artifacts or random noise. They motivate a disagreement-aware framework for unsupervised multi-source mutation preference distillation, where agreement provides useful supervision and disagreement is treated as informative uncertainty rather than simply discarded.

# 5. Method: Disagreement-Aware Preference Distillation

Section 4 shows that mutation signals from ESM2, ESM-IF, and FoldX are informative but systematically heterogeneous. They differ in substitution preferences, distributional sharpness, and cross-method agreement, suggesting that they should not be treated as calibrated observations of a single underlying target. We therefore consider a multi-source unsupervised distillation setting: given several heterogeneous computational mutation sources, we aim to train a student mutation model without experimental mutation labels, source reliability annotations, or cross-source calibration signals.

Under this setting, directly regressing to raw reference scores is ill-defined because different sources operate on different numerical scales and encode different semantic meanings. Instead, our method uses each source through its within-source mutation preference structure. Pairwise preference directions provide scale-invariant supervision, while source-normalized margins provide a local estimate of preference strength within each source. We then aggregate partially consistent preferences through agreement-weighted consensus and regularize the student against overconfident predictions where sources strongly disagree.

## 5.1. Multi-Source Distillation Setup

Given a protein sequence $x$ and a target residue position $i$, the student model $f_\theta$ predicts a 20-dimensional mutation preference score vector

$$S_i = f_\theta(x, i) \in \mathbb{R}^{20}, \qquad (2)$$

where $S_i(a)$ denotes the student score assigned to amino acid $a$ at position $i$. The scores are used only to compare candidate substitutions at the same residue position and are not required to have calibrated probabilistic or physical meaning.

For the same sequence and position, each fixed reference source $k \in \mathcal{K}$ provides a mutation score vector

$$R_i^{(k)} \in \mathbb{R}^{20}, \qquad (3)$$

where $\mathcal{K} = \{\text{ESM2}, \text{ESM-IF}, \text{FoldX}\}$ in our implementation. All reference outputs are fixed during training. The student is initialized from ESM2-650M and fine-tuned using only the computational mutation preferences constructed in Section 3; no experimental mutation labels are used.

## 5.2. Source-Normalized Pairwise Preferences

Because reference scores are not calibrated across sources, we use within-source pairwise preferences as the basic supervision unit. For source $k$, position $i$, and a pair of amino

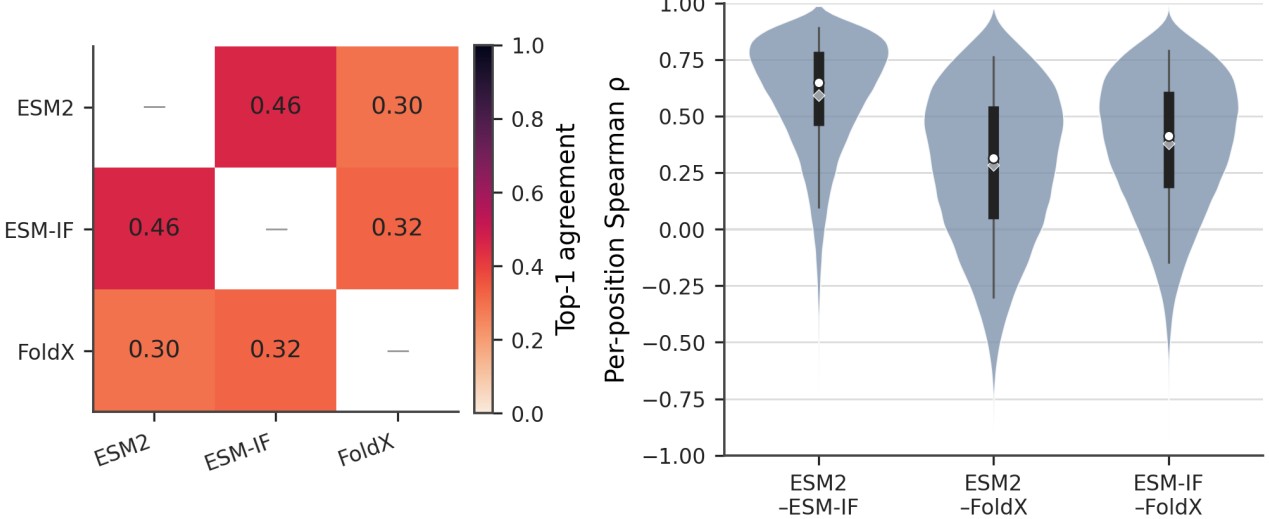

*Figure 5.* **Cross-method agreement and ranking structure. Left**: Top-1 agreement rates. **Right**: per-position Spearman rank correlation. FoldX exhibits low agreement with the other two models, providing a complementary mutation ranking signal.

acids $(a, b)$, we define the reference pairwise margin as

$$\Delta_{i,a,b}^{(k)} = R_i^{(k)}(a) - R_i^{(k)}(b). \quad (4)$$

A positive margin indicates that source $k$ prefers amino acid $a$ over $b$ under the same sequence context.

Although the sign of $\Delta_{i,a,b}^{(k)}$ is invariant to source-specific score scaling, its magnitude is not directly comparable across sources. We therefore normalize pairwise margins within each source:

$$\widetilde{\Delta}_{i,a,b}^{(k)} = \frac{\Delta_{i,a,b}^{(k)}}{\tau_k + \epsilon}, \qquad \tau_k = \mathbb{E}_{i,a,b}\left[\left|\Delta_{i,a,b}^{(k)}\right|\right], \quad (5)$$

where $\tau_k$ is the average absolute pairwise margin of source $k$ over training pairs, and $\epsilon$ is a small constant for numerical stability. This normalization preserves the preference direction within each source while expressing margin size relative to that source's own typical scale.

The corresponding student margin is

$$\Delta_{i,a,b}^{\text{stu}} = S_i(a) - S_i(b). \quad (6)$$

For a preference direction $a \succ b$, we use the logistic pairwise ranking penalty

$$\ell\left(\Delta_{i,a,b}^{\text{stu}}\right) = \log\left(1 + \exp\left(-\Delta_{i,a,b}^{\text{stu}}\right)\right). \quad (7)$$

This penalty serves as the basic building block for multi-source preference supervision.

### 5.3. Agreement-Weighted Consensus Supervision

We next aggregate pairwise preferences across heterogeneous sources. For a candidate preference $a \succ b$, let

$$\mathcal{K}_{i,a,b}^+ = \left\{k \in \mathcal{K} \mid \widetilde{\Delta}_{i,a,b}^{(k)} > 0\right\} \quad (8)$$

denote the set of sources that support this direction. We define the agreement weight as

$$A_{i,a,b} = \frac{\left|\mathcal{K}_{i,a,b}^+\right|}{|\mathcal{K}|}. \quad (9)$$

This term measures the fraction of reference sources that support the same preference direction. We refer to this graded aggregation as soft consensus, because agreement modulates supervision strength continuously rather than determining whether a pair is kept or discarded.

To incorporate source-internal preference strength, we average the normalized positive margins among supporting sources:

$$\bar{\Delta}_{i,a,b}^+ = \frac{1}{\left|\mathcal{K}_{i,a,b}^+\right|} \sum_{k \in \mathcal{K}_{i,a,b}^+} \widetilde{\Delta}_{i,a,b}^{(k)}. \quad (10)$$

The resulting agreement-weighted consensus loss is

$$\mathcal{L}_{\text{soft}} = A_{i,a,b} \bar{\Delta}_{i,a,b}^+ \ell\left(\Delta_{i,a,b}^{\text{stu}}\right). \quad (11)$$

Here, $A_{i,a,b}$ captures cross-source agreement, while $\bar{\Delta}_{i,a,b}^+$ captures the average source-normalized strength of the supporting preferences. Thus, preferences supported by more sources and stronger source-normalized margins provide stronger supervision, whereas partially disputed preferences are retained with reduced weight.

## 5.4. Disagreement-Aware Margin Regularization

Agreement-weighted supervision controls how strongly a preference should be learned, but it does not by itself prevent the student from becoming overly confident in regions where reference sources disagree. As observed in Section 4, different sources can make sharp yet conflicting predictions. We therefore explicitly regularize student confidence according to cross-source disagreement.

For each mutation pair, we quantify disagreement using the variance of source-normalized margins:

$$D_{i,a,b} = \text{Var}_{k \in \mathcal{K}} \left( \widetilde{\Delta}_{i,a,b}^{(k)} \right). \tag{12}$$

A larger value indicates stronger conflict among reference sources regarding the relative preference between $a$ and $b$.

We then penalize excessive student margins on high-disagreement pairs:

$$\mathcal{L}_{\text{dis}} = D_{i,a,b} \max \left( 0, \left| \Delta_{i,a,b}^{\text{stu}} \right| - B \right), \tag{13}$$

where $B$ is a fixed tolerance margin. This term does not force the student to be uncertain everywhere. Instead, it limits unjustified over-separation between candidate mutations when the reference sources strongly disagree. In other words, the model is encouraged to learn confident preferences where sources provide consistent evidence, while avoiding overconfident decisions in regions of high cross-source conflict.

## 5.5. Final Objective

The final training objective combines agreement-weighted preference supervision with disagreement-aware margin regularization:

$$\mathcal{L} = \mathcal{L}_{\text{soft}} + \lambda \mathcal{L}_{\text{dis}}, \tag{14}$$

where $\lambda$ controls the strength of disagreement-aware regularization.

When the disagreement term is removed, the objective reduces to consensus-only distillation. For ablation, we also consider a hard-consensus variant, where a pairwise preference is used only if a majority of reference sources support the same direction, i.e., $|\mathcal{K}_{i,a,b}^+| > |\mathcal{K}|/2$; otherwise, the pair is discarded. In contrast, our soft-consensus objective retains all partially supported preferences and weights them by the agreement fraction $A_{i,a,b}$. In all experiments, these variants share the same student architecture, training data, and optimization settings, so performance differences can be attributed to the design of the distillation objective.

## 6. Experiments

We evaluate the proposed framework on ProteinGym (Notin et al., 2023), a standard benchmark for zero-shot protein mutation effect prediction. Our experiments assess whether heterogeneous mutation signals can be integrated effectively without experimental mutation labels or cross-source score calibration. Specifically, we evaluate three aspects of the framework: (i) whether graded consensus supervision improves over hard consensus filtering, (ii) whether explicit disagreement-aware regularization further improves generalization, and (iii) whether physics-based FoldX augmentation provides complementary information beyond sequence- and structure-derived sources. Together, these experiments test the central claim that agreement and disagreement among heterogeneous computational sources can provide useful supervision for zero-shot mutation effect prediction.

### 6.1. Experimental Setup

**Benchmark.** We adopt the substitution-only benchmarks provided by ProteinGym, which evaluate single amino-acid substitutions across a diverse collection of proteins, functional contexts, and experimental assays.

**Evaluation protocol.** All models are evaluated in a zero-shot manner. No experimental mutation data from ProteinGym are used during training. Performance is measured using macro-averaged Spearman rank correlation, following the official ProteinGym evaluation protocol. We evaluate on all substitution-only datasets whose corresponding protein sequences contain fewer than 1000 amino acids, resulting in a total of 201 datasets used for macro-averaging.

**Training data and supervision.** The student model is obtained by fine-tuning an ESM2-650M backbone on the PDB-wide mutation preference dataset constructed in Section 3. Supervision signals are derived solely from reference mutation models, including sequence-based protein language models, structure-conditioned inverse folding models, and physics-based energy functions. No experimental fitness, stability, or $\Delta\Delta G$ labels are used at any stage of training.

**Baselines** We compare against sequence-only baselines, individual information-source reference models, dataset ablations, and objective variants; detailed baseline descriptions are provided in Appendix D.

Unless otherwise specified, all experiments share the same student architecture, training data, sampling strategy, and optimization hyperparameters. Performance differences can therefore be directly attributed to the design of the training objective.

### 6.2. Main Results

Table 1 reports macro-averaged Spearman correlations on ProteinGym. The disagreement-aware variant achieves the best overall performance among all evaluated meth-

| Method | Overall | Activity | Binding | Expression | Fitness | Stability |
|---|---|---|---|---|---|---|
| *Information-source reference models* | | | | | | |
| ESM-IF | 0.459 | 0.412 | 0.381 | 0.437 | 0.347 | 0.624 |
| FoldX (reference) | 0.424 | 0.350 | 0.304 | 0.368 | 0.322 | 0.608 |
| *Sequence-only baselines* | | | | | | |
| VespaG | 0.487 | **0.496** | 0.363 | 0.461 | **0.465** | 0.533 |
| GEMME | 0.480 | 0.485 | 0.378 | 0.447 | 0.465 | 0.519 |
| VESPA | 0.467 | 0.476 | 0.365 | 0.409 | 0.462 | 0.500 |
| ESM2-650M | 0.446 | 0.433 | 0.356 | 0.440 | 0.396 | 0.523 |
| *Dataset ablations* | | | | | | |
| ESM-IF aug only | 0.480 | 0.447 | 0.371 | 0.478 | 0.433 | 0.568 |
| *Variants of the proposed method* | | | | | | |
| Hard consensus | 0.468 | 0.439 | 0.363 | 0.467 | 0.394 | 0.582 |
| Soft consensus | 0.475 | 0.448 | 0.375 | 0.473 | 0.401 | 0.586 |
| Disagreement-aware ($\lambda = 0.2$) | **0.498** | 0.467 | **0.389** | **0.495** | 0.414 | **0.625** |

*Table 1.* Macro-averaged Spearman correlation on ProteinGym substitution benchmarks. The disagreement-aware variant, trained with FoldX-augmented multi-source mutation preferences, achieves the best overall performance.

ods, outperforming individual reference sources, sequence-only baselines, and consensus-only variants. It also obtains the strongest results on Binding, Expression, and Stability, while remaining competitive on Activity and Fitness.

Among objective variants, performance improves from hard consensus to soft consensus and further to disagreement-aware distillation, supporting the benefit of graded agreement modeling and explicit cross-source disagreement regularization. The ESM-IF-only dataset ablation, which excludes FoldX supervision, obtains a lower overall correlation ($\rho \approx 0.48$), suggesting that physics-based energy signals provide complementary information beyond sequence- and structure-derived sources.

### 6.3. Ablation Studies

We conduct ablation studies to assess the contribution of key components in the proposed framework. Specifically, we examine the effects of consensus formulation (hard versus soft consensus) and the impact of disagreement-aware regularization strength. These ablations show that soft consensus provides better coverage and more effective utilization of heterogeneous supervision than hard consensus, and that explicit regularization prevents the student from becoming unjustifiably confident in high-disagreement regions. Additional ablation results and sensitivity analyses are reported in Appendix A.

## 7. Conclusion

We introduce a large-scale, multi-source mutation preference atlas that aligns sequence-based, structure-conditioned, and physics-based mutation signals across experimentally resolved protein structures. Our analysis shows that these sources encode systematically different preferences and

agreement structures, motivating a disagreement-aware distillation framework that integrates heterogeneous signals without experimental mutation labels or cross-source score calibration. On ProteinGym, the resulting model achieves the best overall zero-shot mutation effect prediction among evaluated methods. Beyond these results, we expect the PDB-wide atlas to serve as a useful resource for protein engineering foundation models, pretraining objectives, and preference-based learning frameworks.

## Impact Statement

This paper presents work whose goal is to advance the field of machine learning for protein modeling and mutation effect analysis. While protein engineering technologies may have broad applications in biotechnology and medicine, the methods and datasets introduced in this work are intended for research use and do not raise specific ethical concerns beyond those commonly associated with advances in machine learning.

## Acknowledgements

Research was supported in part by the Molecule Maker Lab Institute: An AI Research Institutes program supported by NSF under Award No. 2505932, and the DOE Center for Advanced Bioenergy and Bioproducts Innovation (U.S. Department of Energy, Office of Science, Biological and Environmental Research Program under Award Number DE-SC0018420). Any opinions, findings, and conclusions or recommendations expressed in this publication are those of the author(s) and do not necessarily reflect the views of the U.S. Department of Energy.

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

| Method | Overall | Activity | Binding | Expression | Org. Fitness | Stability |
|---|---|---|---|---|---|---|
| Hard consensus | 0.468 | 0.439 | 0.363 | 0.467 | 0.394 | 0.582 |
| Soft consensus | 0.475 | 0.448 | 0.375 | 0.473 | 0.401 | 0.586 |
| Disagreement-aware ($\lambda = 0$) | 0.475 | 0.448 | 0.375 | 0.473 | 0.401 | 0.586 |
| Disagreement-aware ($\lambda = 0.01$) | 0.479 | 0.447 | 0.371 | 0.485 | 0.406 | 0.594 |
| Disagreement-aware ($\lambda = 0.1$) | 0.495 | 0.466 | 0.386 | 0.493 | 0.413 | 0.620 |
| Disagreement-aware ($\lambda = 0.2$) | **0.498** | **0.467** | **0.389** | 0.495 | **0.414** | **0.625** |
| Disagreement-aware ($\lambda = 0.5$) | 0.493 | 0.462 | 0.375 | **0.496** | 0.411 | 0.622 |
| Disagreement-aware ($\lambda = 1.0$) | 0.495 | 0.465 | 0.378 | 0.493 | 0.413 | 0.619 |

*Table 2.* Comparison of consensus-based baselines and disagreement-aware distillation under different disagreement strengths $\lambda$, evaluated on ProteinGym. Results are reported as macro-averaged Spearman correlation coefficients, both overall and broken down by task category.

## A. Consensus and Disagreement-Aware Ablation

We investigate the effect of different consensus formulations and the strength of disagreement-aware regularization on zero-shot mutation effect prediction. All variants are trained using the same backbone, training data, and optimization settings, and differ only in the design of the training objective.

Table 2 shows that modeling disagreement explicitly leads to consistent performance improvements over both hard and soft consensus baselines. Soft consensus outperforms hard consensus by weighting supervision according to cross-source agreement, but remains inferior to disagreement-aware distillation.

Varying the regularization strength $\lambda$ reveals a clear performance trend. Moderate disagreement-aware regularization ($\lambda \in [0.1, 0.5]$) yields the strongest results across tasks, with $\lambda = 0.2$ achieving the best overall performance. In contrast, overly weak regularization ($\lambda = 0.01$) leads to degraded performance, while completely disabling disagreement modeling ($\lambda = 0$) also results in substantially lower correlations.

These results indicate that cross-source disagreement carries informative uncertainty signals rather than noise. Ignoring disagreement, as in hard or soft consensus distillation, leads to inferior generalization, whereas explicitly modeling disagreement with appropriate strength improves robustness and performance in zero-shot mutation effect prediction.

## B. Additional Mutation Landscape Analyses

This appendix provides supplementary analyses that support and extend the findings in Section 4 of the main text. All results are based on the same unified mutation preference representation, where each residue position is associated with a 20-amino-acid mutation distribution.

### B.1. Additional Analyses of Wild-type Self-preference

In addition to the decision-level Top-1 WT self-preference reported in the main text (Fig. 2), we further analyze WT preservation using two complementary statistics: (i) the mean predicted probability mass assigned to the WT amino acid, and (ii) the fraction of positions where WT appears among the Top-3 mutations.

As shown in Fig. 6, these distribution-level and relaxed decision-level metrics exhibit trends consistent with the Top-1 analysis in the main text, while providing a more nuanced view of WT competitiveness under less strict decision criteria.

### B.2. Global Mutant Preference Independent of WT Identity

To further characterize WT-independent biases in mutation space, we analyze the global frequency with which each amino acid is selected as the Top-1 mutation across all residue positions.

As shown in Fig. 7, FoldX exhibits a pronounced bias toward small and conformationally flexible residues, whereas ESM2 and ESM-IF display different global preference profiles.

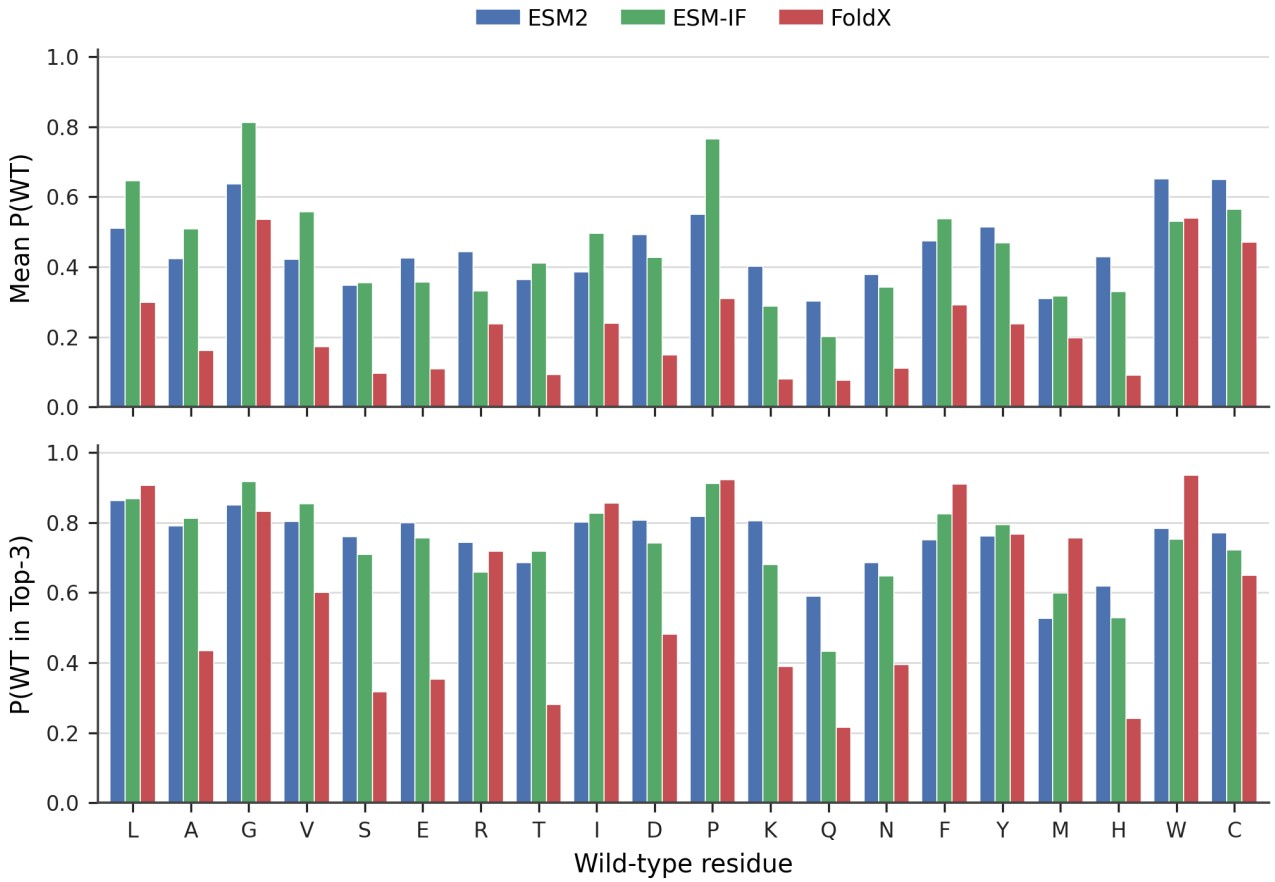

*Figure 6.* **Additional WT self-preference statistics. Top**: mean predicted probability assigned to the WT amino acid. **Bottom**: fraction of positions where WT appears among the Top-3 mutations. Results are grouped by WT residue type.

### B.3. Detailed WT→Mutant Substitution Matrices

We provide full WT→Mutant substitution matrices to complement the qualitative discussion in the main text (Section 4). As shown in Fig. 8, these matrices illustrate substantial differences in numerical scale and substitution behavior across methods.

### B.4. Additional Confidence and Uncertainty Analyses

As a complement to the entropy analysis in Section 4.2 of the main text, we report the distribution of Top-1−Top-2 logit gaps across residue positions.

As shown in Fig. 9, FoldX exhibits consistently smaller decision margins, whereas ESM2 and ESM-IF show larger gaps at a subset of positions.

### B.5. Relaxed Cross-method Agreement via Top-*k* Overlap

Beyond strict Top-1 agreement, we analyze relaxed agreement using Top-$k$ overlap, measured by Jaccard similarity for $k = 3$ and $k = 5$.

As shown in Fig. 10, agreement remains limited even under relaxed criteria, consistent with the ranking-based analyses in the main text.

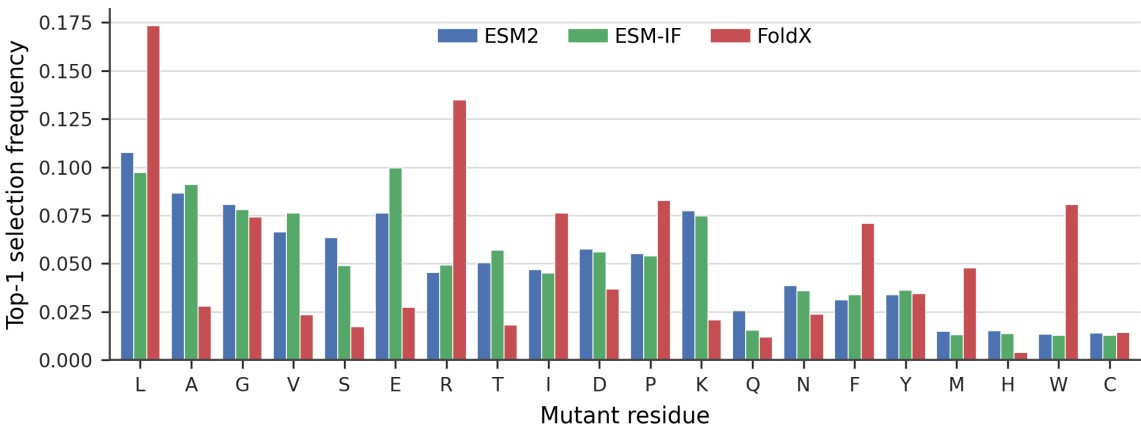

*Figure 7.* **Global mutant preference across methods.** Frequency with which each amino acid is selected as the Top-1 mutation, aggregated across all residue positions.

### B.6. Entropy–Disagreement Relationship

We further analyze the relationship between per-position uncertainty and cross-method disagreement. While entropy correlates with Top-1 disagreement to some extent, we observe a substantial number of residue positions with low entropy but high disagreement.

As illustrated in Fig. 11, these cases indicate conflicts driven by differing inductive biases rather than uncertainty alone.

## C. FoldX Mutation Evaluation Details

FoldX (Schymkowitz et al., 2005) is an empirical, physics-inspired energy function widely used for estimating the structural stability effects of protein mutations. In this work, FoldX is employed to evaluate single-site mutations on experimentally resolved protein structures.

For each protein structure, we first apply standard structure preparation procedures provided by FoldX to ensure a consistent energetic baseline. Single-site mutations are then introduced by replacing the side chain at the target residue position with the mutated amino acid. Following mutation, FoldX performs local structural repacking to relax side-chain conformations in the vicinity of the mutation site. The energetic effect of a mutation is quantified as the difference in predicted folding free energy between the mutant and the corresponding wild-type structure.

The resulting energy difference, denoted as $\Delta\Delta G$, reflects the relative energetic favorability of a mutation under the FoldX energy model. In this work, these energy differences are not treated as ground-truth fitness measurements, but are instead used as structure-aware mutation signals that inform relative mutation preferences.

## D. Baselines

We compare the proposed approach against three categories of baselines that represent the dominant paradigms for zero-shot mutation effect prediction.

**Sequence-only baselines.** We include state-of-the-art sequence-based methods VespaG (Marquet et al.) and GEMME (Laine et al., 2019) that operate solely on protein sequences without explicit structural or physical information. These baselines represent the strongest existing approaches based on evolutionary and language-model-derived signals.

**Information-source reference models.** We further compare against representative structure-conditioned and physics-based models, including an inverse folding model (ESM-IF) and a physics-based energy function (FoldX). These models provide complementary mutation signals derived from structural constraints and physical stability considerations, respectively.

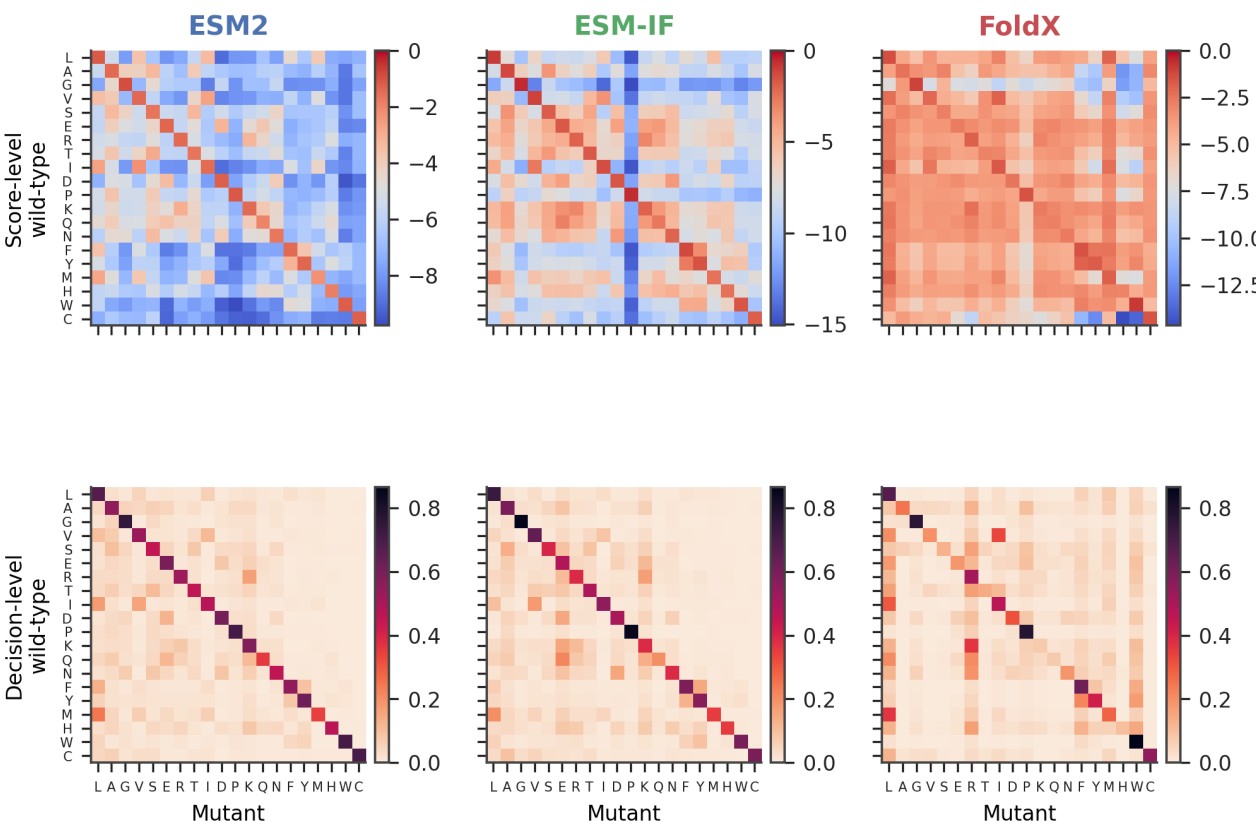

*Figure 8.* **WT→Mutant substitution landscapes (full matrices). Top**: average WT→Mutant logit matrix (score-level). **Bottom**: WT→Mutant Top-1 optimality matrix (decision-level).

**Variants of the proposed method.**    To assess the effect of different supervision formulations, we report results for multiple variants of our approach, including hard-consensus distillation, soft-consensus distillation, and the full disagreement-aware formulation. All variants share the same backbone architecture and training data, and differ only in the design of the training objective.

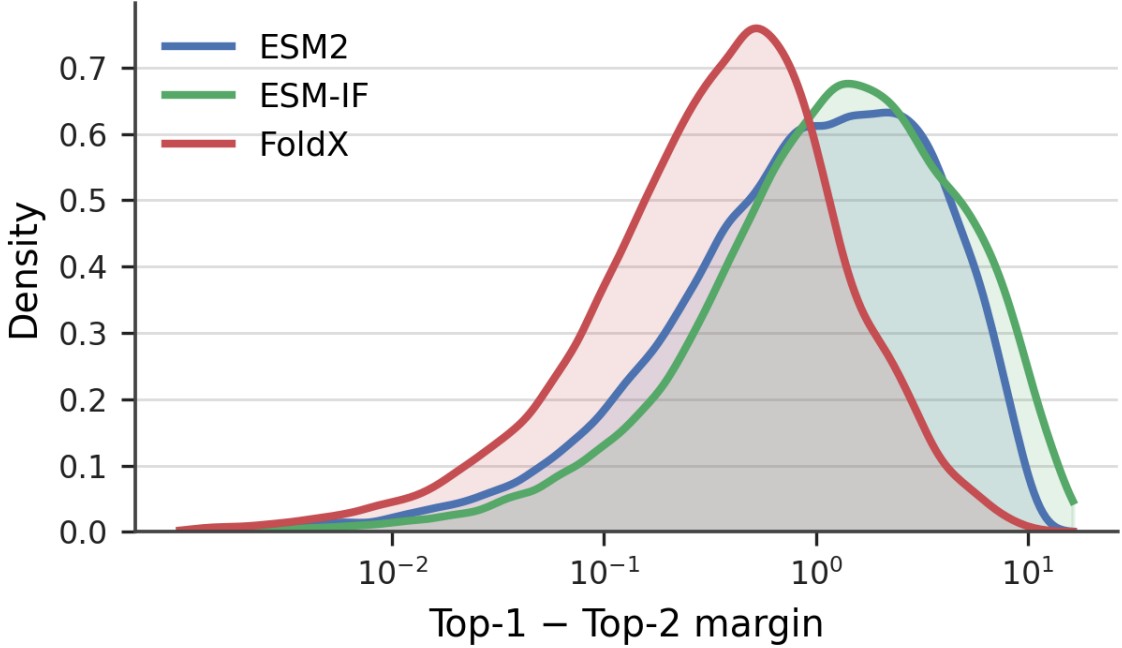

*Figure 9.* **Top-1−Top-2 logit gap distributions.**

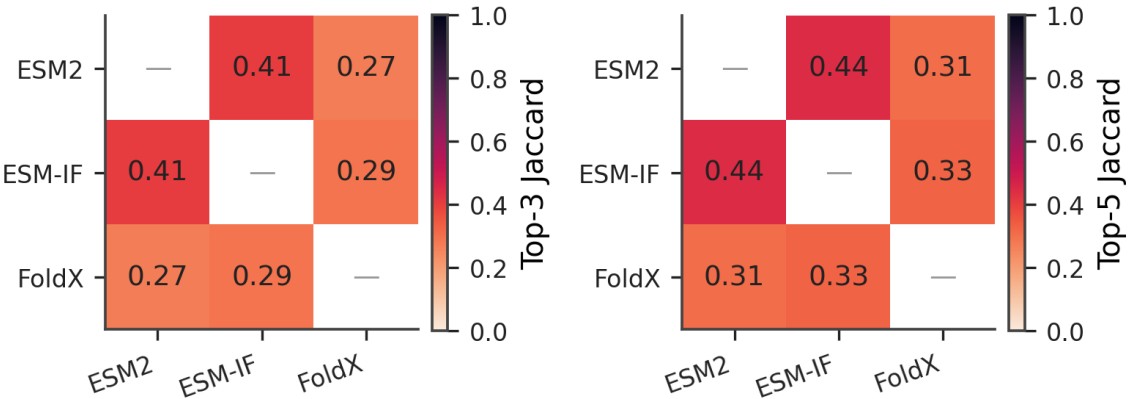

*Figure 10.* **Top-*k* overlap between mutation predictions.**

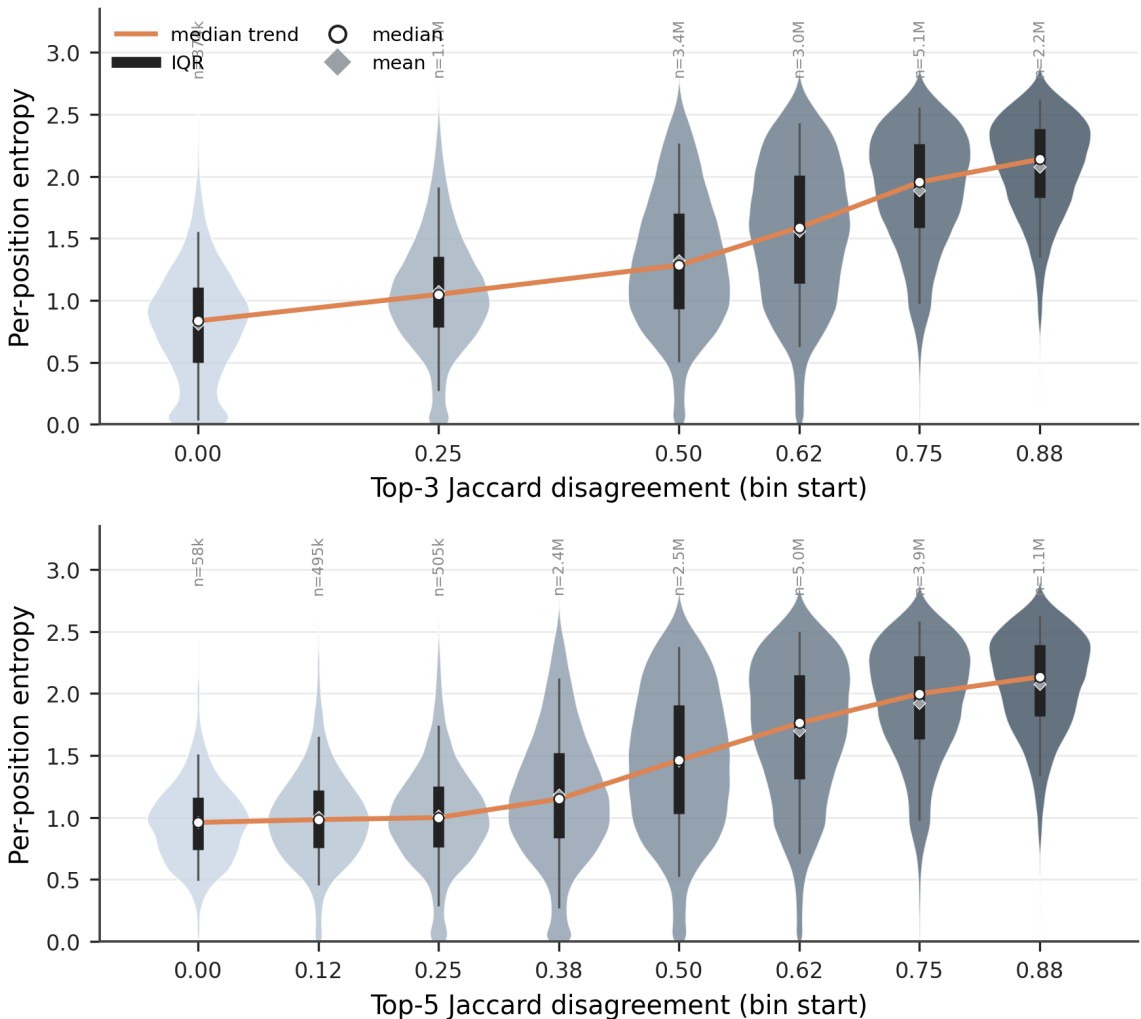

*Figure 11.* **Entropy vs. cross-method disagreement.** Relationship between mean per-position entropy and disagreement under $k = 3$ (top) and $k = 5$ (bottom).

