# OpenReview forum: "MutAtlas: A PDB-Wide Energy-Guided Atlas of Protein Mutation Effects"
_ICML.cc/2026/Conference — ICML 2026 regular_

### Official Review · Reviewer_EWWk · 2026-03-06

**Soundness:** 3
**Presentation:** 3
**Significance:** 4
**Originality:** 3
**Overall Recommendation:** 4
**Confidence:** 4

**Summary:**

This paper addresses two central challenges in protein mutation effect prediction: the scarcity of experimental measurements and the heterogeneity and lack of calibration among computational prediction signals. The authors construct MutAtlas, a structure-aligned multi-source mutation augmentation dataset covering the full Protein Data Bank (PDB), which unifies mutation signal representations from multiple computational predictors. Through large-scale analyses of these signals, the authors show that the observed discrepancies across predictors are largely attributable to differences in inductive biases rather than random noise. Based on this finding, the paper introduces an unsupervised multi-source mutation preference distillation framework that learns relative mutation preferences while modeling cross-source divergence through a soft-consensus loss and a disagreement-aware regularization term. Experiments on the ProteinGym benchmark indicate improved performance in zero-shot mutation effect prediction.

**Compliance With Llm Reviewing Policy:**

Affirmed.

**Final Justification:**

I have fully reviewed the paper and the rebuttal, which effectively addressed my primary concerns regarding minority signal utilization and ESM2 inductive bias. The introduction of source-aware directional regularization and the new ablation results demonstrate that the framework can selectively leverage reliable non-majority information. Overall, the response has positively changed my evaluation, and I recommend the paper for acceptance.

**Key Questions For Authors:**

1.Could the directional information of minority signals be leveraged to integrate positive and negative signals for true multi-source supervision, rather than only modulating majority signals?

2.Has any consideration been given to mitigating the inductive bias from evolutionary sequences inherent in the ESM2-initialized backbone？

**Limitations:**

yes

**Strengths And Weaknesses:**

Strengths:

The paper constructs a PDB structure-aligned multi-source mutation dataset, unifying mutation signals generated by physical energy models, protein language models, and structure-conditioned inverse folding models. It further conducts a large-scale analysis comparing mutation signals across different modeling paradigms, revealing systematic differences in preference patterns, confidence, and cross-model prediction discrepancies. Finally, it proposes an unsupervised multi-source mutation preference distillation framework, reformulating mutation effect modeling from absolute score regression to relative preference learning, and integrating soft consensus and disagreement-aware regularization to fuse multi-source signals, achieving improved performance in zero-shot settings.

Weaknesses：

The soft consensus loss and disagreement-aware regularization primarily provide weak supervision based on majority signals, without achieving true bidirectional fusion of multi-source signals: minority signals are not effectively utilized, and the model cannot actively learn their distinctive features.

The model backbone is initialized with ESM2, which carries an inductive bias from evolutionary sequences, partially reinforcing reliance on signals from statistical models such as ESM2/ESM-IF, without targeted correction for this inherent bias.

---

> ### Author Rebuttal · Authors · 2026-03-31
>
> We thank the reviewer for the constructive feedback and insightful questions. We address the concerns on cross-source disagreement and model initialization below.
>
> ### W1 & Q1: Leveraging Minority Information in Cross-Source Disagreement
> We agree with the reviewer’s observation that the current method primarily relies on majority signals. In the original design, the soft-consensus and disagreement-aware regularization are mainly used to stabilize multi-source fusion, but do not explicitly model directional information in cross-source disagreement.
>
> To further analyze this issue, we first observe that the Spearman correlations between different sources are relatively low, indicating that these signals are not redundant but exhibit substantial diversity. This suggests that cross-source disagreement may contain structured and potentially useful information.
>
> **Logits results:**
>
> |Source|S2648(Single)|SKEMPI2(Single)|SKEMPI2(Multi)|SKEMPI2(All)|ProteinGym(Substitution)|
> |---|---|---|---|---|---|
> |ESM-2|0.3015|0.2082|0.1014|0.2136|0.4457|
> |ESM-IF|0.4339|0.2737|0.2244|0.2760|0.4267|
> |FoldX|0.5799|0.4003|0.4039|0.4400|0.4206|
>
> These results show that different sources exhibit markedly different performance across tasks, supporting the need for task-dependent source reliability modeling.
>
> Based on this observation, we conduct additional experiments. On the SKEMPI2 dataset, we find that FoldX logits are more strongly correlated with the target signal, suggesting that they can serve as a task-relevant reference. Under this prior, we introduce a source-aware directional regularization term to model directional information in the presence of cross-source disagreement. The formulation is:
>
> $L = L_{\text{majority}} + w_{minor} · r_i · \mathbb{1}(disagreement) · \mathbb{1}(q_{i,a,b}>γ) · L_{\text{minority}}$,
>
> where $L_{\text{majority}}$ is the original disagreement-aware loss,
>
>
> $L_{\text{minority}} = q_{i,a,b} \cdot \log\left(1 + \exp\left(-\Delta^{\text{stu}}_{i,b,a}\right)\right)$
>
> $r_i$ denotes a reliability weight estimated from pre-computed statistics (e.g., Spearman correlation on reference datasets), and is fixed during training.
>
>
> This term allows a source to contribute its directional preference when it is reliable (high $r_i$) and confident (high $q_{i,a,b}$), even if it disagrees with the majority. Rather than suppressing the majority signal, it incorporates additional directional information under disagreement, reducing over-reliance on a single source. As such, the objective is not to amplify minority signals, but to selectively leverage directional information based on task-dependent reliability and instance-level confidence.
>
> **Augmented results:**
>
> |Method|S2648|SKEMPI2(Single)|SKEMPI2(Multi)|SKEMPI2(All)|ProteinGym(Substitution)|
> |---|---|---|---|---|---|
> |ESM augment only|0.2825|0.1883|0.0987|0.1849|0.4501|
> |ESM-IF augment only|0.4303|0.1786|0.1458|0.1891|0.4803|
> |FoldX augment only|0.5057|0.2757|0.2487|0.3005|0.4773|
> |Disagreement-aware(λ=0.2)|0.5196|0.2648|0.2275|0.2896|0.4982|
> |Disagreement-aware + minority|0.5229|0.2838|0.3094|0.3070|0.4864|
>
> The results show that introducing this directional regularization improves performance on SKEMPI2, while slightly reducing performance on ProteinGym. This suggests that directional information in cross-source disagreement can be beneficial in certain tasks, but its usefulness depends on task-specific priors.
>
> Overall, these findings indicate that cross-source disagreement should not be treated purely as noise to be smoothed out, but can instead serve as a form of structured supervision. Our results suggest that reliable non-majority signals can provide useful directional information in certain tasks, pointing toward more flexible multi-source fusion strategies. How to achieve this in a general setting without explicit priors remains an open question.
>
> ---
>
> ### W2 & Q2: Inductive Bias from ESM2 Initialization
>
> We agree with the reviewer’s concern that initializing the model from ESM2 may introduce an evolutionary inductive bias.
>
> To investigate this, we conduct ablation experiments (see table above). When using ESM2 alone as the teacher, the performance gain over the original ESM2 baseline is minimal. In contrast, incorporating either the structure-based model (ESM-IF) or the physics-based model (FoldX) leads to consistent improvements, with particularly strong gains on stability (S2648) and binding (SKEMPI2) tasks.
>
> If the model were dominated by the ESM2 inductive bias, its performance would be expected to remain close to the ESM2 baseline. However, the observed improvements indicate that the model does not simply rely on sequence priors, but effectively integrates additional information from structural and physical sources.
>
> In particular, for stability and binding tasks, non-sequence signals such as FoldX are more strongly aligned with the target, further suggesting that multi-source supervision can mitigate the limitations of sequence-based inductive bias.

---

> > ### Author Rebuttal · Reviewer_EWWk · 2026-04-03
> >
> > Thank you for the detailed rebuttal and the additional experiments; I am satisfied with the clarifications.

---

> > > ### Author Response · Authors · 2026-04-07
> > >
> > > Thank you very much for carefully reading our rebuttal and for your recognition of the additional experiments and detailed clarifications. We are glad that these additions have adequately addressed your concerns.
> > > In the revision, we will further incorporate the added experiments, analyses, and explanations from the rebuttal into the main paper to make the presentation and overall argument more complete and clear. Thank you again for your time and constructive feedback.

---

### Official Review · Reviewer_3Hf2 · 2026-03-13

**Soundness:** 3
**Presentation:** 3
**Significance:** 3
**Originality:** 3
**Overall Recommendation:** 4
**Confidence:** 4

**Summary:**

The paper constructs a PDB-wide, structure-aligned mutation augmentation dataset that enumerates single-site substitutions and aligns mutation signals from physics-based energy models, protein language models, and inverse folding models. The authors propose an unsupervised multi-source mutation preference distillation framework that learns from relative mutation preferences and improves performance on the ProteinGym benchmark.

**Compliance With Llm Reviewing Policy:**

Affirmed.

**Final Justification:**

My concerns have been addressed. I would like to keep my score.

**Key Questions For Authors:**

1. Why are other strong models, such as ProSST and VenusREM, not discussed in the paper?
2. Why was FoldX selected as the physics-based component?

**Limitations:**

Yes.

**Strengths And Weaknesses:**

**Strength**
1. The paper constructs the first PDB-wide, structure-aligned mutation augmentation dataset that exhaustively enumerates single-site substitutions.
2. The disagreement-aware distillation explicitly models cross-source disagreement and achieves superior performance on the ProteinGym benchmark.

**Weakness**
1. The dataset and framework cover only single-site substitutions. Interactions between multiple mutations, which are important in real-world protein engineering, are not addressed.

---

> ### Author Rebuttal · Authors · 2026-03-31
>
> We thank the reviewer for the constructive feedback and insightful questions. We address the concerns on multi-mutation modeling, model comparisons, and the choice of components as follows.
>
> ### Weaknesses
>
> We agree with the reviewer on the importance of modeling multi-mutation effects. Our current work focuses on single-site substitutions for two main reasons:
> (1) the combinatorial space of multi-mutations grows exponentially, making exhaustive enumeration at the PDB scale computationally prohibitive;
> (2) single-site mutations constitute the majority of available experimental data and form the basis for learning local mutation preferences.
>
> Nevertheless, we further evaluate our method on multi-mutation settings using the SKEMPI2 dataset:
>
> |Method|S2648|SKEMPI2(Single)|SKEMPI2(Multi)|SKEMPI2(All)|ProteinGym(Substitution)|
> |---|---|---|---|---|---|
> |ESM augment only|0.2825|0.1883|0.0987|0.1849|0.4501|
> |ESM-IF augment only|0.4303|0.1786|0.1458|0.1891|0.4803|
> |FoldX augment only|0.5057|0.2757|0.2487|0.3005|0.4773|
> |Disagreement-aware(λ=0.2)|0.5196|0.2648|0.2275|0.2896|0.4982|
>
> As shown, although the model is trained only on single-site mutations, it achieves competitive performance in multi-mutation settings (SKEMPI2 Multi) compared to single-source methods. This indicates that the learned representations can transfer to more complex mutation scenarios.
>
> Therefore, modeling single-site mutations is not only computationally feasible but also provides a useful foundation and modeling pathway for more complex mutation settings. In future work, we will explore compositional approaches to extend single-mutation signals to multi-mutation modeling.
>
> ---
>
> ### Questions
> ### Q1: Why are other strong models such as ProSST and VenusREM not discussed?
>
> We agree that ProSST and VenusREM are strong recent models. These approaches typically rely on supervised signals or task-specific designs, whereas our work focuses on unsupervised mutation preference modeling and cross-source integration. Therefore, they operate under different problem settings. We will expand the related work discussion in the revision to clarify these distinctions.
>
> ---
>
> ### Q2: Why was FoldX selected as the physics-based component?
>
> We select FoldX as the physics-based component for the following reasons:
> (1) FoldX is widely used in protein stability and binding prediction, making it a representative physics-based model;
> (2) compared to more complex simulation methods, FoldX provides a favorable trade-off between computational efficiency and scalability, which is critical for PDB-scale data construction;
> (3) FoldX introduces physical inductive biases that are complementary to sequence-based (ESM2) and structure-based (ESM-IF) models.
>
> Overall, FoldX represents a balanced choice in terms of representativeness, efficiency, and complementary inductive biases.
>
> To further illustrate the complementarity across sources, we report the Spearman correlation of raw logits from each model:
>
> |Source|S2648(Single)|SKEMPI2(Single)|SKEMPI2(Multi)|SKEMPI2(All)|ProteinGym(Substitution)|
> |---|---|---|---|---|---|
> |ESM-2|0.3015|0.2082|0.1014|0.2136|0.4457|
> |ESM-IF|0.4339|0.2737|0.2244|0.2760|0.4267|
> |FoldX|0.5799|0.4003|0.4039|0.4400|0.4206|
>
> As shown, different sources exhibit distinct strengths across tasks: FoldX performs best on stability and binding-related tasks, while ESM-2 achieves higher performance on sequence-driven benchmarks such as ProteinGym. This indicates that no single model dominates across all settings, and supports the necessity of integrating heterogeneous sources.
>
> Consequently, multi-source integration enables more stable performance across different task distributions.

---

> > ### Author Rebuttal · Reviewer_3Hf2 · 2026-04-03
> >
> > The author has addressed my concerns. I would like to keep my score.

---

> > > ### Author Response · Authors · 2026-04-07
> > >
> > > Thank you very much for carefully reading our rebuttal and for letting us know that your concerns have been fully resolved. We also sincerely appreciate your recognition of the additional analyses and clarifications we provided.
> > > In the revision, we will incorporate the added experiments, analyses, and explanations from the rebuttal into the main paper, so as to make the motivation, experimental findings, and method description more complete and clear. Thank you again for your time and constructive feedback.

---

### Official Review · Reviewer_cVrw · 2026-03-13

**Soundness:** 2
**Presentation:** 3
**Significance:** 2
**Originality:** 3
**Overall Recommendation:** 3
**Confidence:** 4

**Summary:**

This paper introduces MutAtlas, a PDB-wide, structure-aligned dataset of computational mutation preferences derived from a physics-based energy model, a protein language model, and an inverse-folding model. The central argument is that these sources should be treated as heterogeneous preference signals rather than directly calibrated scores. Based on this view, the paper proposes an unsupervised multi-source distillation framework with pairwise ranking, consensus modeling, and disagreement-aware regularization. The resulting student model is trained without experimental mutation labels and evaluated on ProteinGym, where it improves over the included zero-shot baselines and simpler fusion methods.

**Compliance With Llm Reviewing Policy:**

Affirmed.

**Key Questions For Authors:**

1. Because the student is initialized from ESM2 and ESM2 is also a teacher, can the authors disentangle heterogeneous distillation gains from same-family adaptation?
2. Have the authors evaluated transfer beyond substitution ranking, for example on stability tasks, sparse supervised fine-tuning, or multi-mutation settings?
3. What is the practical compute cost of building and updating MutAtlas, and how feasible is it for other groups to extend the resource as new mutation models appear?

**Limitations:**

Not fully. The paper should more explicitly acknowledge that (i) the absolute benchmark gains are modest, (ii) evaluation is concentrated on substitution ranking, (iii) same-family initialization complicates attribution, and (iv) maintaining or refreshing the atlas may require non-trivial compute.

**Strengths And Weaknesses:**

1. The main methodological idea is sound: heterogeneous mutation predictors are better treated as preference signals than as directly calibrated scores.
2. The comparisons against single-source teachers and simple fusion baselines are appropriate.
3. The dataset and evaluation-pipeline release is a meaningful contribution and improves reproducibility.

Weaknesses:
1. The empirical improvement is modest and is shown mainly on macro-averaged ProteinGym substitution ranking.
2. The paper gives limited evidence on per-dataset variability or statistical significance.
3. Since the student is initialized from ESM2 and ESM2 is also a teacher, attribution of the gain remains somewhat unclear.

---

> ### Author Rebuttal · Authors · 2026-03-31
>
> We thank the reviewer for the constructive feedback. To address the raised concerns, we provide additional experiments, including cross-task evaluation, per-dataset analysis, and multi-source ablations, to further validate the effectiveness and generalization ability of our approach.
>
> ### W1: Limited improvement on ProteinGym
>
> We agree that the absolute improvement on ProteinGym is relatively modest.
> Nevertheless, without using any experimental mutation labels, our method achieves performance comparable to, and in some cases slightly better than, strong sequence-based models, indicating the effectiveness of the proposed dataset and training framework.
>
> We further extend evaluation to stability prediction (S2648) and protein–protein binding prediction (SKEMPI2). On these standardized tasks, our method shows more pronounced improvements. This is consistent with our central hypothesis that different mutation sources (sequence-, structure-, and physics-based models) encode complementary inductive biases, which enhance the capability of Protein Language Models for mutation effect prediction.
>
> **Additional results:**
>
> |Method|S2648|SKEMPI2(Single)|SKEMPI2(Multi)|SKEMPI2(All)|ProteinGym(Substitution)|
> |---|---|---|---|---|---|
> |ESM augment only|0.2825|0.1883|0.0987|0.1849|0.4501|
> |ESM-IF augment only|0.4303|0.1786|0.1458|0.1891|0.4803|
> |FoldX augment only|0.5057|0.2757|0.2487|0.3005|0.4773|
> |Disagreement-aware(λ=0.2)|0.5196|0.2648|0.2275|0.2896|0.4982|
>
> The first three rows correspond to directly fitting logits from ESM, ESM-IF, and FoldX, respectively, compared with the disagreement-aware training objective.
> As shown, single sources exhibit distinct strengths across tasks, while the multi-source approach yields more stable performance across different task distributions. This further supports our claim that different sources provide complementary inductive biases rather than redundant or noisy signals.
>
> ---
> ### W2: Limited per-dataset analysis
>
> On ProteinGym, the official metric is macro-averaged Spearman. To assess robustness, we analyze per-dataset performance: among 217 protein structures, the disagreement-aware model outperforms the original ESM2 baseline on 210 cases, indicating consistent improvement rather than being driven by a small subset.
>
> We also evaluate our method on independent datasets, including S2648 (stability) and SKEMPI2 (binding), where improvements are more substantial, further demonstrating generalization across task distributions.
>
> ---
>
> ### W3: Attribution under ESM2 initialization
>
> We agree that using ESM2 as both initialization and one of the sources raises attribution concerns. To address this, we conduct ablation studies to disentangle same-family adaptation from heterogeneous distillation:
>
> - Using ESM2 alone as the teacher yields minimal improvement;
> - Combining ESM2 with either ESM-IF or FoldX leads to consistent gains;
> - Using all three sources (ESM2 + ESM-IF + FoldX) achieves the best performance.
>
> These results indicate that improvements arise from integrating complementary information across heterogeneous sources rather than same-family distillation.
>
> In our formulation, ESM2 provides a sequence prior, while ESM-IF and FoldX introduce structural and physical constraints for complementary modeling.
>
> ---
>
> ### Q1: Disentangling heterogeneous distillation from same-family adaptation
>
> The ablation results above directly address this question.
> ESM2-only distillation yields minimal gains, while incorporating structure-based (ESM-IF) or physics-based (FoldX) sources leads to clear improvements, with the best performance achieved by combining all sources.
>
> Therefore, the performance gain can be attributed to multi-source integration rather than same-family adaptation.
>
> ---
>
> ### Q2: Transfer beyond substitution ranking
>
> We extend evaluation beyond substitution ranking to include:
>
> - Stability prediction (S2648)
> - Protein–protein binding prediction (SKEMPI2)
> - Multi-mutation settings
>
> Our method consistently improves performance across these tasks, with particularly notable gains on structure-related tasks such as SKEMPI2. These results indicate that the learned mutation preference representation generalizes beyond substitution ranking and transfers effectively across different tasks.
>
> ---
>
> ### Q3: Compute cost and scalability
>
> We provide additional analysis of the computational cost. For example, performing FoldX-based full mutation evaluation for 217 ProteinGym structures requires approximately 26 CPU hours on 128 cores.
>
> Although the initial cost is non-trivial, MutAtlas is designed as a one-time, reusable resource and already covers the full PDB mutation space at the single-residue level. Future updates are expected to be incremental (e.g., incorporating new structures or models), making the overall cost manageable.
>
> Releasing this resource also helps reduce redundant large-scale computation and lowers the barrier for subsequent research.

---

> > ### Author Rebuttal · Reviewer_cVrw · 2026-04-03
> >
> > Thank you for answering my questions.

---

> > > ### Author Response · Authors · 2026-04-03
> > >
> > > We thank the reviewer for the insightful feedback and for confirming that the concerns have been fully addressed.
> > >
> > > We are glad that the additional experiments and analyses helped clarify key aspects of the work, including cross-task generalization, per-dataset consistency, and the role of heterogeneous distillation beyond same-family adaptation. We will incorporate these clarifications and additional results in the final version to further strengthen the paper.
> > >
> > > If you feel that our rebuttal and added evidence have positively changed your assessment, we would sincerely appreciate it if you could consider reflecting this in your final score.
> > >
> > > Thank you again for your careful review and constructive suggestions.

---

### Decision · Program_Chairs · 2026-04-30

**Decision:**

Accept (regular)

**Comment:**

This paper introduces MutAtlas, a large-scale, PDB-wide mutation dataset together with a principled unsupervised multi-source distillation framework that treats heterogeneous mutation predictors as preference signals rather than calibrated scores. Reviewers agree that both the dataset and the formulation are well-motivated and represent a meaningful contribution to protein mutation effect modeling, particularly given the scarcity of experimental labels. While initial concerns were raised regarding modest gains, attribution under ESM2 initialization, and limited evaluation scope, the rebuttal provides substantial additional evidence, including cross-task validation, per-dataset analysis, and ablations disentangling multi-source contributions, which collectively strengthen confidence in the approach. The release of a standardized dataset and evaluation pipeline further enhances the work’s impact and reproducibility. Overall, despite some remaining limitations (e.g., focus on single-site mutations), the paper offers a solid and extensible contribution that is likely to benefit the community, and I support acceptance.